# Predictive Relevance Uncertainty for Recommendation System

## ABSTRACT

Click-through Rate (CTR) module is the foundation block of recommendation system and used for search, content selection, advertising, video streaming etc. CTR is modelled as a classification problem and extensive research is done to improve the CTR models. However, uncertainty method for these models are still an unexplored area. In this work we analyse popular uncertainty methods in the context of recommendation system. We found that popular uncertainty models fails to capture the predictive uncertainty of the CTR model that exist unique to the recommendation models and is not prevalent in the traditional classification models. We empirical show why a different uncertainty measure is required for the recommendation system CTR prediction models. We propose PRU (Predictive Relevance Uncertainty), a single forward pass uncertainty approach for a sample as a distance from the predictive relevance samples of the training data. We show the efficacy of the proposed predictive relevance uncertainty (PRU) on selective prediction. Further, we demonstrate the utility of the proposed framework on the downstream task of OOD detection and active learning while maintaining the latency of a single pass deterministic model.

**ACM Reference Format:**
Anonymous Author(s). 2023. Predictive Relevance Uncertainty for Recommendation System. In *Proceedings of ACM Conference (Conference'17)*. ACM, New York, NY, USA, 9 pages. https://doi.org/10.1145/nnnnnnn.nnnnnnn

## 1 INTRODUCTION

Click-through Rate (CTR) prediction problem is ubiquitous in today's e-commerce, advertising, search and video streaming services. CTR models predict the likelihood of a user clicking on an item, be it a product, web article or an ad. The modeling involves two steps: in the first step (known as inference), one mines short-term and long-term history of the user and item metadata to rank all eligible items and surfaces the one (or few) with the highest estimated CTR. In the second step (referred to as training), the system collects appropriate feedback based on customer's interaction and retrains the model with the latest available information.

CTR prediction is challenging for multiple reasons. First, due to relatively rare occurrence of positive samples, the training data is often insufficient to fit large parameter space of the model which leads to variability in its predictions. Moreover, dynamic user behavior, new customers and items, external events may require the model to predict on distribution of samples that was not observed during

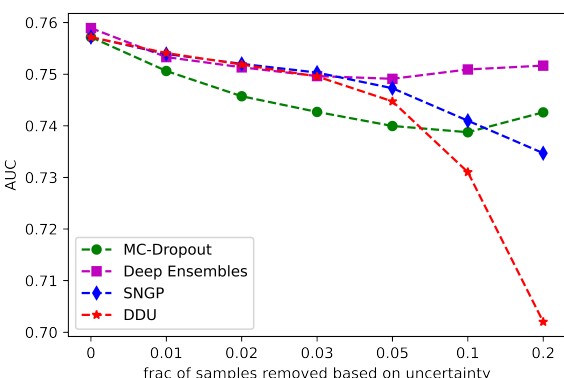

**Figure 1: Uncertainty experiment on the Avazu dataset using MC-Dropout, DDU, SNGP and Deep Ensembles. We report AUC after discarding the least confident predictions from the test dataset.**

training time. Finally, CTR prediction involves out-of-distribution (OOD) samples by virtue of positional and presentation bias, content selection bias and user targeting. This results in inaccurate and over-confident estimation of CTR leading to a drop in its performance.

One approach is to identify OOD samples via uncertainty quantification and filter them out at prediction time. Uncertainty captures the notion of model's confidence in accurately predicting the target label and therefore datapoints with high uncertainty are typically associated with erroneous predictions. Uncertainty estimates can benefit the model several ways e.g. it allows to make informed decisions and allocate resources wisely. Uncertainty is the deciding factor to trade exploration (recommending new items) with exploitation (recommending popular items) in multi-arm and contextual bandit algorithms. Finally, uncertainty estimates can guide active learning strategies, enabling the system to focus on uncertain predictions and acquire new data to improve the model. However unlike NLP, computer vision where much progress has been made on uncertainty-aware learning, reliable and efficient estimation of uncertainty is an open problem for recommender systems.

Click through rate estimation is often modeled as a classification problem where the goal is to classify an input into two classes: clicked and non-clicked. We expect the model to be inaccurate on highly uncertain points and hence if we were to remove these low-confidence predictions, the model performance would improve on the rest. In Figure 1, we plot the result of this experiment on the Avazu [5] dataset using state of the art uncertainty quantification techniques: DDU [21], SNGP [20], MC-Dropout [9] and Deep Ensembles [17]. We see a reverse trend where instead of improvement, the performance either stays flat or degrades significantly as we

filter out least confident predictions. This suggests that uncertainty for CTR models is poorly explained by the current SOTA literature and needs deeper investigation. Recommendation system setting is different from the traditional classification setting as in the recommendation setup, neighbourhood of a datapoint has heterogeneous labels suggesting high degree of overlap between class-conditionals. Fig 2 shows the class conditionals for two recommendation data-set.

In this work, we investigate uncertainty estimation for CTR models and recommendation systems in general. We first provide insights on why recommendation problems need special treatment for uncertainty quantification. Guided by our insights, we propose PRU (Predictive relevance Uncertainty), a novel single pass deterministic uncertainty model that can be utilised over any existing CTR model for uncertainty estimation with no changes to the model architecture. Essentially, PRU is a meta-learning algorithm where we can plug in any model for CTR estimation and get accurate and efficient estimation of uncertainty. Experiments on benchmark datasets show superiority of PRU over state of the art baselines for uncertainty estimation in recommendation domains across variety of downstream tasks. We make the following contribution in this paper:

- We empirically study the SOTA uncertainty quantification for recommender systems and show that they fail to capture the true notion of predictive uncertainty.
- We present Predictive Relevance Uncertainty (PRU), a novel approach to quantify uncertainty for deep CTR prediction models, which can provide efficient uncertainty estimations along with the predictions and is compatible with any deep CTR models.
- We evaluate the effectiveness of PRU on selective prediction, out of distribution (OOD) detection and active learning. We perform a thorough and comprehensive set of experiments on three public benchmark datasets for CTR modeling comparing against several SOTA techniques for uncertainty estimation.
- Our experimental result suggests that PRU achieves statistically significant +16% lift in selective prediction as compared to the strong uncertainty baselines. To highlight the accuracy of uncertainty quantification, we evaluate PRU on the downstream task of out of distribution (OOD) detection and active learning. Compared to strong baselines, PRU achieves +5% lift in OOD detection and +0.9% , +7% lift in active learning for different datasets.

## 2 RELATED WORKS

### 2.1 CTR prediction Problem

The purpose of CTR prediction is to estimate the probability that a user will click on an item. Although loosely used in the context of click, the definition is broad enough to capture any interaction such as purchase, video stream etc. One challenge in recommender systems is to find balance between *memorization* and *generalization*. Memorization refers to learning frequent co-occurrence of items from historical data whereas generalization refers the ability of the model to predict on unseen patterns. Cheng et al [4] proposed Wide&Deep which combines a DNN with a linear model and are trained jointly. The linear model encodes sparse features

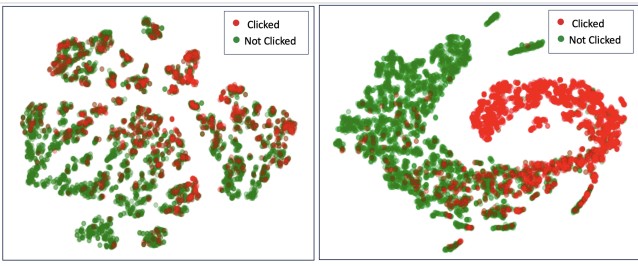

**Figure 2: Feature distribution of clicked (in red) and non-clicked (in green) samples for Avazu (left) and MovieLens (right) dataset (best viewed in color).**

such as item-id, cross features between user and item that helps in memorization. On the other hand, the DNN component helps in generalizing to unseen patterns.

DeepFM [11] is another popular technique for CTR estimation that augments a traditional Factorization Machine (FM [23]) with a DNN component. Unlike Wide&Deep, DeepFM can be trained end to end without any feature engineering. DeepFM has further been extended to incorporate explicit feature interactions [19], adding a diversity loss in training objective to avoid overfititng [3] etc.

Zhou et al [33] presents Deep Interest Network (DIN) which adaptively learns user representation based on historical behavior with respect to certain ad. Despite learning contextual representation of users, DIN offers limited support for feature interaction. This was subsequently addressed in Deep&Cross networks (DCN [29, 30]).

### 2.2 Uncertainty

Popular methods of quantifying uncertainty includes Bayesian Neural Network [2], MC Dropout [9] and Deep Ensembles [18]. MC Dropout is a scalable alternative to the BNN models [9]. Deep ensemble aggregates collection of trained neural network to quantify the model uncertainty while MC dropout uses dropout enabled forward pass to quantify the model uncertainty. These framework can captures the model uncertainty where the model uncertainty is quantified by a sample distance from the decision boundary. Also, these are computationally expensive as it requires multiple forward passes to obtain the uncertainty measure. Therefore, single pass deterministic models are proposed where the uncertainty of a sample is quantified based on the distance/density from the training data[20, 21, 28].

Predictive uncertainty can be classified into two kinds [8, 13]: *aleatoric* or data uncertainty and *epistemic* or model uncertainty. While epistemic uncertainty can be reduced by collecting more data, aleatoric uncertainty, on the other hand requires instrumenting new features. Recent research has focused on disentangling uncertainty into these two components. Disentanglement is helpful for classification scenario where decision can be refused or delayed like as in a classifier with reject option [1] or reducing uncertainty in active learning scenario [25]. There has been recent work on disentangling uncertainty. For example, Kendall et al [14] define a model to estimate both aleatoric and epistemic uncertainty for regression and classification models. Matias et al [26] extends this techinque to richer class of models.

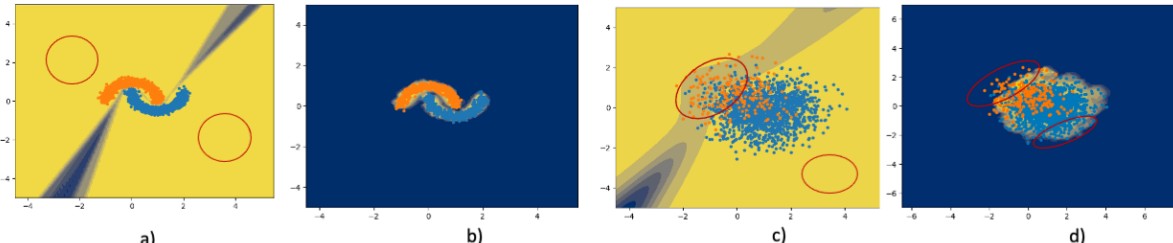

**Figure 3: Failure modes for Model uncertainty (Deep ensembles) and Density/Distance aware uncertainty (DDU) in two moons dataset and Synthetic data-set with class overlap and imbalance. Blue denotes the high uncertainty region and yellow denotes the low uncertainty region. Red circle denotes the failure points. a) Model uncertainty for two moons dataset b) Density/Distance aware uncertainty (DDU) for two moons dataset. c) Model uncertainty for synthetic data d) Density/Distance aware uncertainty (DDU) for synthetic data**

There has been extensive study of uncertainty models in the classification [16, 24], computer vision [15] and NLP [7, 22, 31] domain but remain unexplored in the recommendation literature. One recent work on recommendation system uncertainty is evaluated only for Movielens and Netflix dataset where user-item pairs are used to quantify different form of uncertainty [6]. Further, it doesn't quantify uncertainty based on the distance from the training data distribution. In this work, we propose PRU where the uncertainty can be quantified on any CTR dataset and model architecture.

## 3 UNCERTAINTY FOR RECOMMENDATION PROBLEMS

We argue that uncertainty should be higher for 1) OOD (Out of Distribution) samples i.e. for the samples that are away from the training data distribution. As the model has not seen this type of data samples at the training time, prediction on them cannot be trusted, 2) points near the decision boundary: as the model is confused about which class the samples belongs to, it leads to high variance in the model score. We check the notion of uncertainty for the traditional classification setting of two-moons dataset [1] and class overlap with imbalance synthetic dataset in Fig 3.

We plot the uncertainty surface for the two moons dataset in Fig 3a for model uncertainty using deep ensemble [18] and Fig 3b for density/distance aware uncertainty (DDU [21]). Yellow denotes the low uncertainty region and blue denotes the high uncertainty region. We show the failure modes for the uncertainty algorithm using red circles in Fig 3. For example in Fig 3a, we observe lower uncertainty in the red circle, whereas it is expected to be higher as this region (OOD sample region) is away from the training data distribution, therefore it is a failure mode for the uncertainty algorithm. Distance aware uncertainty captures the OOD detection problem better as compared to the model uncertainty as shown in Fig 3b. We observe high uncertainty for both OOD samples and decision boundary for density/distance aware uncertainty (DDU) in the traditional classification setting of two-moons dataset as shown in Fig 3b.

We then simulate the behavior of class overlap and imbalance observed in the recommendation system by using noise of 0.7 and class imbalance ratio of 0.2 on the two moons dataset. We observe this setting resembles the recommendation system setting as there is class overlap (neighbourhood of an instance is not homogeneously populated by instances of the same true class) and class imbalance (observe negative class ( non clicks) dominating the positive class (clicks)) as shown in Fig 3c and 3d. In this setting model and distance aware uncertainty fails to capture the correct notion of uncertainty.

We observe high uncertainty for the minority class as the decision boundary for the model gets shifted towards the minority class in case of model uncertainty. Further, OOD samples (points away from the training data distribution) will be considered as low uncertainty samples as shown in Fig 3c. This notion of uncertainty can be harmful to the recommendation system, as the in distribution minority class samples are assigned higher uncertainty then the OOD samples. Using density/distance-aware uncertainty in this setting also fails to provide a true notion of uncertainty. This is because the uncertainty will be lowest in the class overlapping region, given that the training data distribution is concentrated mostly for both classes in that region. Furthermore, uncertainty will be lowest for the predictive relevance samples (samples where the model is confident in clicks/No clicks). Although density/distance aware uncertainty assign high uncertainty to the points away from the training data distribution. But it also assign high uncertainty to the predictive relevance samples where the model precision is high. This can again hurt the performance when utilizing uncertainty for the downstream task in recommendation system.

## 4 PREDICTIVE RELEVANCE UNCERTAINTY

In this work we propose, Predictive Relevance Uncertainty (PRU) to estimate the uncertainty of deep CTR prediction models, which can provide efficient uncertainty quantification and is compatible with any deep CTR model. We define the notion of predictive uncertainty as a distance from the training samples that are highly predictive relevant. Highly predictive relevant are the samples where the model precision is high. In this way, the proposed framework PRU will capture the uncertainty of the OOD samples that are away from the training data distribution as well as points that have high variance

---

[1]https://scikit-learn.org/stable/modules/generated/sklearn.datasets.make_moons.html

**Figure 4: High-level architecture of the model and training steps.**

due to the class overlap issue. At high level, PRU implements the following steps: 1) identify training samples that have high predictive relevance, 2) fit a density estimator on the regularized feature space of predictive relevant samples, 3) estimate uncertainty of a test sample by computing likelihood under the density estimator. Next we describe each of these steps in detail.

## 4.1 Predictive Relevant Instance Selection

We define predictive relevant samples as the ones where model precision is high, thereby, predicted score and the ground truth labels are close. Model will have lower loss on the predictive relevance samples. We fit a Gaussian Mixture Model (GMM) to cluster datapoints on their observed training loss i.e. points with similar loss are put in the same cluster. Samples belonging to the GMM component with the smallest mean can be selected as high predictive relevant samples. This modeling procedure however is class-agnostic, and reflect the same degree of loss for both the classes (click/no click). Given that in recommender systems, majority of the data is biased towards the negative class (i.e. no click), this approach will end up picking the predicted relevant samples primarily from the majority class. To avoid this behavior, we use class specific GMM to determine the predictive relevant samples for each class individually.

## 4.2 Density Estimation

A deep learning CTR model is typically composed of a feature transformation layer $h(x)$ that maps input instances to a hidden representation space and an output function $g(h(x))$ that maps the hidden representations to an output space. To utilize the feature space for distance awareness and density estimation, the hidden representation space is required to follow the bi-Lipschitz constraint so that distance in the latent space $d_h(h(x), h(x'))$ is bounded by the distance $d_x(x, x')$ in the input data manifold, for any inputs $x, x'$. More formally, we require $h(\cdot)$ to satisfy the bi-Lipschitz condition [20],

$$L * d_x(x, x') \leq d_h(h(x), h(x')) \leq U * d_x(x, x') \quad (1)$$

for positive and bounded constants $0 < L < 1 < U$. Here $d_x$ and $d_h$ denote any meaningful distance metric in the input and hidden representation space. The upper Lipschitz bound $d_h(h(x), h(x')) \leq U * d_x(x, x')$ is an important requirement to ensure robustness of the feature transformation layer which prevents the hidden representation $h(x)$ to be overly sensitive to perturbations in the input

space. On the other hand, the lower Lipschitz bound $L * d_x(x, x') \leq d_h(h(x), h(x'))$ prevents the hidden representations to be unnecessarily invariant to large changes in the input space. Together, the bi-Lipschitz condition ensures that distances in the representation space are truthful representation of distances in the input space.

---

**Algorithm 1** PRU Density Estimation

---

1: **procedure** TRAIN
2:     Train Baseline DNN model $p(\mathbf{y}|\mathbf{f}_\theta(\mathbf{x}))$ with $(\mathbf{X}, \mathbf{Y})$
3:     Get predictive relevance samples $\mathbf{X_{pr}} \subset \mathbf{X}$
4:     **for** each class $c$ with samples $\mathbf{X_c} \subset \mathbf{X_{pr}}$ **do**
5:         compute feature representation $z(x) = h_\theta(x)$
6:         Compute mean $\mu_c$:
7:         $\mu_c \leftarrow \frac{1}{|\mathbf{X_c}|} \sum_{\mathbf{x} \in \mathbf{X_c}} h_\theta(x)$
8:         Compute covariance $\Sigma_c$:
9:         $\Sigma_c \leftarrow \frac{1}{|\mathbf{X_c}|-1} \sum_{\mathbf{x} \in \mathbf{X_c}} (h_\theta(x) - \mu_c) \cdot (h_\theta(x) - \mu_c)^T$
10:       Compute Cluster weights $\pi_c$:
11:       $\pi_c \leftarrow \frac{|\mathbf{X_c}|}{|\mathbf{X_{pr}}|}$
12:       $q(z|y = c) \sim \mathcal{N}(\mu_c; \Sigma_c), q(y = c) = \pi_c$

---

Gradient penalty and spectral normalization are two techniques of enforce the bi-Lipschitz condition [27]. Spectral normalization is a simpler technique used in distance aware uncertainty frameworks. It is applied to hidden weights in order to enforce bi-Lipschitz smoothness in representations [20, 21] Techniques proposed in [20, 21] restrict the framework to ResNet [12] to ensure sensitivity to the change in input. To utilize the existing architectures, we use two side gradient penalty, regularising the Jacobian with respect to the input embedding of the model. Therefore, the proposed PRU framework requires no changes in the model architecture and thereby compatible to any CTR framework. Overall we use spectral normalization to ensure stabilized training and two-sided Jacobian regularisation to encourage sensitivity to the inputs as shown in Fig 4.

Post-training, we fit a Gaussian mixture model on the loss residuals i.e. $\log f(x)$ or $\log(1 - f(x))$ depending on the true label. The GMM returns $m$ mixing components $\{(\mu_i, \Sigma_i)\}_{i=1 \cdots m}$ and a vector of mixing coefficient $\pi_k$ for each sample $x_k$. Datapoints are mapped to the cluster based on max probability in $\pi_i$ and we pick the cluster with lowest mean as the relevant sample set $\mathbf{X_{pr}}$.

As a final step, we identify the positive and negative samples in $\mathbf{X_{pr}}$. For each class $c$, let $\mathbf{X_c} \subset \mathbf{X_{pr}}$ be the subset of relevant samples. We fit a multi-variate normal distribution $q(z \mid y = c) = \mathcal{N}(z \mid \mu_c, \Sigma_c)$ via maximum likelihood estimation. Algorithm 1 highlights the pseudocode of our algorithm.

## 4.3 Uncertainty Quantification using PRU

We quantify uncertainty by calculating the marginal likelihood of the hidden feature representation under density estimator on the regularized predictive relevance samples as $q(z) = \sum_{y=c} q(z|y = c) \cdot q(y = c)$

*4.3.1 Epistemic and Aleatoric Uncertainty:* PRU captures both epistemic and aleotoric uncertainty similar to other distance aware uncertainty frameworks [20, 28]. When a point is far from the training data distribution (epistemic uncertainty), PRU uncertainty will

be high as the likelihood of belonging to any of the class will be low. When a point lies in class overlapping data distribution (aleotoric uncertainty), PRU uncertainty will be high as these samples will be far from the predicted relevance samples of the classes.

## 5 EXPERIMENTS

In this section, we will evaluate the efficacy of the proposed framework. First, we will assess the uncertainty estimates of the proposed framework in the traditional CTR setting to determine the level of trust we can place in the model predictions. Then, we will evaluate how the proposed framework would perform if utilized for the downstream tasks of OOD detection and active learning. We evaluate the performance of the proposed Predictive Relevance Uncertainty (PRU) with the following SOTA uncertainty quantification methods.

- MC-Dropout[10] : MC-Dropout provides a distribution over predictions via a sequence of forward passes by dropout-enabled pre-trained network. We use 10 forward passes and compute the variance in predictions to get the uncertainty estimates.
- DEEP ENSEMBLES[18]: DEEP ENSEMBLES aggregates a collection of trained neural networks with different initialization. We use 5-ensemble models to compute the uncertainty estimates.
- Spectral-Normalized Neural Gaussian Process (SNGP) [20]: SNGP is a single-pass, deterministic model that encode predictive uncertainty via distance-awareness.
- Deep Deterministic Uncertainty (DDU) [21] : DDU first fit each class component by computing the empirical mean and covariance, of each class feature vectors. Then compute the likelihood of a test sample belonging to each class component to quantify confidence.

### 5.1 Relevance to Prediction Scores

We use the following Evaluation Metrics to evaluate the uncertainty estimates.

- Selective prediction : High uncertainty is linked with low prediction performance. Therefore, the obtained confidence scores are thresholded and model is only evaluated on low uncertainty samples. Since we are filtering high uncertainty samples, it is expected that on the remaining samples, the model performance metric will improve. We report the increase/ decrease in AUC and PRAUC after filtering the high uncertainty samples.
- Latency: We report the time in (ms) to compute the uncertainty for 1k samples (ms/1k examples).

*5.1.1 Datasets.* We experiment with two open benchmark CTR datasets, 1) **MovieLens:** [2] Movielens data contains tagging record (user ID, movie ID, tag). Tag denotes the target and is assigned class 1 if user tags the movie. 2) **Avazu:**[3] This dataset contains 22 feature fields including user features and advertisement features for mobile advertisements and uses click records by users as the labels. We reuse the preprocessed data by [5] and follow the same settings

---

[2]https://grouplens.org/datasets/movielens/
[3]https://www.kaggle.com/c/avazu-ctr-prediction

on data splitting and preprocessing. Further details on creating the data split and preprocessing is provided by BARS [34].

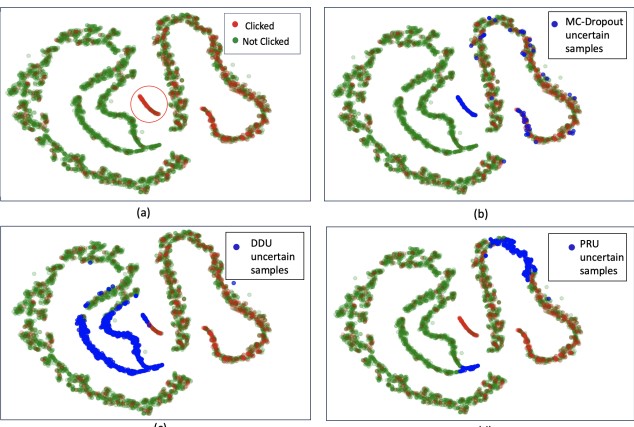

**Figure 5: TSNE plots of the feature extractor layer from the baseline model for Avazu. a) denotes the distribution of features mapped to class: Not clicked (green) and class: Clicked (Red). b) High Uncertainty samples from MC dropout in blue. c) High Uncertainty samples from DDU in blue. d) High Uncertainty samples from PRU (proposed) in blue.**

*5.1.2 Implementation:* We use the two popular backbone model (DeepFM and Wide&Deep) based on FuxiCTR [34, 35]. We set the embedding dimension to 16, default MLP size to [200, 200], learning rate 0.01 to train the backbone model. We use the batch size of 8192 and 20480 for MovieLens and Avazu respectively. We use 10 forward passes for the MC-Dropout, 5 ensemble model for Deep Ensembles. We evaluate PRU for ($m = 3, 4, 5$) gaussian mixture components. Note that we fit the $m$ GMM modes on the training loss to determine the predictive relevant samples.

*5.1.3 Experimentation.* We report AUC metric at different coverage in Table 1. AUC-95 denotes the bps (increase/decrease) in AUC if 5% of the data samples are removed from the evaluation set based on high uncertainty i.e (AUC after filtering - AUC without filtering) × 10000. We report the AUC at different coverage of 95, 90, 80 by filtering 5%, 10% and 20% of the most uncertain data based on the uncertainty algorithm. AUC metric can be biased to the majority data, therefore we also report the PRAUC-90. While the baseline algorithms observe a drop in majority of cases. PRU outperforms and improves the AUC on selective prediction for all the scenarios. The performance of PRU is consistent across modes for the Movielens, but for Avazu we observe a drop for higher modes. As we keep on increasing the modes, the sampled data for density estimator decreases and if we fit higher modes, the density estimator lose out the useful training data distribution. MC Dropout improves on the Movielens dataset, but the performance of MC Dropout significantly drops for Avazu dataset. The drop is even higher for the PRAUC-90 in Avazu. To understand the trend, we plot the TSNE feature distribution of Avazu from the feature extractor layer in Fig 5. Fig 5a represents the class distribution for clicks/not clicked. We observe overlap and imbalance in the data. The samples inside

**Table 1: Selective prediction at different coverage. (\*) indicates the statistically significant of PRU improvement over the best baseline (two-sided t-test with $p < 0.05$)**

| | | DeepFM | | | | | Wide&Deep | | | | |
|---|---|---|---|---|---|---|---|---|---|---|---|
| | | AUC-95 | AUC-90 | AUC-80 | PRAUC-90 | latency | AUC-95 | AUC-90 | AUC-80 | PRAUC-90 | latency |
| Movielens | MC Dropout | 25.63 | 38.47 | 49.21 | 53.90 | 70.23 | 23.97 | 34.65 | 41.34 | 46.05 | 67.87 |
| | 5-ensemble | 7.69 | -1.51 | -37.19 | -33.42 | 35.39 | 10.32 | 8.40 | -12.38 | -12.74 | 36.74 |
| | DDU | -38.19 | -41.81 | -25.86 | -83.09 | 7.22 | -40.20 | -49.33 | -35.52 | -103.34 | 6.99 |
| | SNGP | -21.91 | -37.20 | -58.71 | -65.71 | 18.20 | -20.72 | -35.06 | -55.09 | -56.55 | 15.54 |
| | PRU (m=3) | 28.84 | 47.86 | 66.97 | 77.01 | 7.36 | 29.55 | 48.59 | 66.53 | 77.36 | 7.02 |
| | PRU (m=4) | **29.81\*** | **49.32\*** | **68.03\*** | **79.89\*** | 7.43 | **30.66\*** | **50.64\*** | **68.20\*** | **81.16\*** | 6.98 |
| | PRU (m=5) | 29.28 | 48.45 | 67.49 | 78.23 | 7.23 | 30.01 | 49.41 | 67.32 | 78.90 | 7.12 |
| Avazu | MC Dropout | -144.19 | -134.92 | -67.62 | -774.35 | 107.69 | -150.24 | -146.55 | -91.82 | -805.23 | 103.35 |
| | 5-ensemble | -58.49 | -39.50 | -8.82 | -359.42 | 64.68 | -98.34 | -80.25 | -12.79 | -544.92 | 60.45 |
| | DDU | -115.72 | -246.89 | -534.00 | -196.55 | 16.47 | -122.66 | -258.80 | -546.25 | -275.46 | 14.91 |
| | SNGP | -62.63 | -108.43 | -172.99 | -50.63 | 28.78 | -64.55 | -106.17 | -155.34 | -82.22 | 28.35 |
| | PRU (m=3) | **23.28\*** | **73.34\*** | **146.16\*** | **30.19\*** | 16.25 | **27.95\*** | **72.54\*** | 132.02 | 14.65 | 15.01 |
| | PRU (m=4) | 9.24 | 41.14 | 101.28 | 7.03 | 15.91 | 10.28 | 55.49 | 113.67 | 12.05 | 14.84 |
| | PRU (m=5) | 15.90 | 59.75 | 124.59 | 21.54 | 16.21 | 23.28 | 69.40 | **132.30\*** | **15.06\*** | 14.96 |

the red circle denotes the higher confident samples for the minority class (click). MC-Dropout flags all the highly confident samples of the minority class as uncertain, therefore suffers the highest drop in the PRAUC. DDU flags confident samples from the minority as well as minority as uncertain, while PRU flags the majority of the points in the overlapping region as uncertain. Therefore, PRU is able to achieve the improved performance while maintaining the latency of a single pass backbone model.

## 5.2 OOD Detection

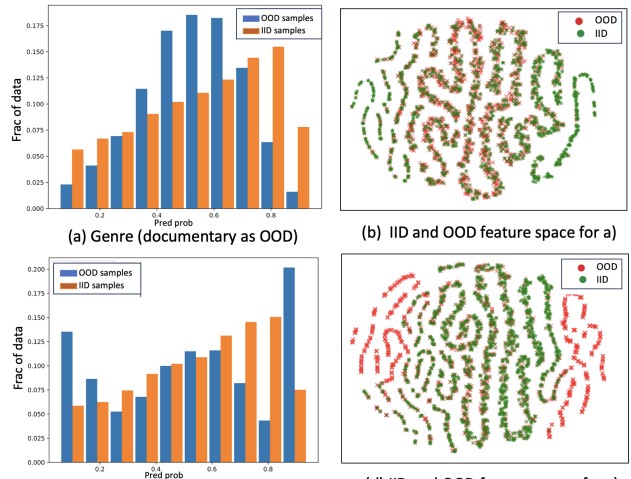

(a) Genre (documentary as OOD)

(b) IID and OOD feature space for a)

(c) Genre (documentary as OOD ) (e+$\eta$)

(d) IID and OOD feature space for c)

**Figure 6: Feature distribution for the two settings used to create the OOD samples**

For the OOD detection task, we expect OOD data samples (data distribution on which the model is not trained) to have higher uncertainty compared to the IID samples (data distribution on which

the model is trained). We label OOD samples as class 1 and IID samples as class 0, and report the AUC score based on the uncertainty estimates. If the uncertainty is high, the sample is expected to be OOD.

*5.2.1 Datasets:* We use category information to define the OOD samples. Datasets defined in section 5.1.1 either doesn't contain category information or it is anonymized. Therefore, we use the following datasets for the OOD detection task setup.

**MovieLens-1M** [4] :The data consists of 1 million movie ranking instances over thousands of movies and users. Each movie has features including its title, year of release, and genres. Titles and genres are lists of tokens. Each user has features including the user's ID, age, gender, and occupation. We transform ratings into binary (The ratings at least 4 are turned into 1 and the others are turned into 0).

**Taobao Display Ad Click** [5] : It contains 1,140,000 users from the website of Taobao for 8 days of ad display / click logs (26 million records). Each ad can be seen as an item in our paper, with features including its ad ID, category ID, campaign ID, brand ID, Advertiser ID. Each user has 9 categorical attributes: user ID, Micro group ID, cms-group-id, gender, age, consumption grade, shopping depth, occupation, city level.

We use the same terminology and processing steps used in [32] to pre-process the MovieLens-1M and Taobao Ad datasets.

*5.2.2 Experimentation:* We use two settings to create the OOD samples. In the first setting, we hold out a subset of data based on category information (e.g., genre in MovieLens-1M and category-id in Taobao Ad). We choose the genre and categories based on their frequency in the data. For MovieLens-1M, we select the least frequent genre as Genre#1 OOD set, corresponding to the documentary genre, and pick the three least frequent genres as Genre#3 OOD set. For Taobao Ad, we pick the bottom 29 category-ids based

---

[4]1 http://www.grouplens.org/datasets/movielens
[5]https://tianchi.aliyun.com/dataset/56

**Table 2: Overall Performance comparison with uncertainty baselines. AUROC on the OOD/IID detection (Backbone model: Wide&Deep). (\*) indicates the statistically significant improvement over the best baseline (two-sided t-test with $p < 0.05$)**

|  | | MovieLens-1M | | Taobao Ad | |
|---|---|---|---|---|---|
|  | Genre#1 | Genre#1 $(e + \eta)$ | Genre#3 | Genre#3 $(e + \eta)$ | Category_id | Category_id $(e + \eta)$ |
| MC Dropout | 0.5608 | 0.4633 | 0.5695 | 0.4744 | 0.5255 | 0.4630 |
| 5-ensemble | 0.5122 | 0.4597 | 0.5353 | 0.4977 | 0.4207 | 0.4680 |
| DDU | 0.3432 | 0.4680 | 0.3480 | 0.4658 | 0.4976 | 0.6018 |
| SNGP | 0.5317 | 0.5667 | 0.4573 | 0.5887 | 0.5782 | 0.5950 |
| PRU (m=3) | 0.5710 | **0.6192**\* | 0.6028 | 0.6088 | 0.5958 | 0.6255 |
| PRU (m=5) | **0.6170** \* | 0.6171 | **0.6096** \* | **0.6226** \* | **0.6537**\* | **0.6346**\* |

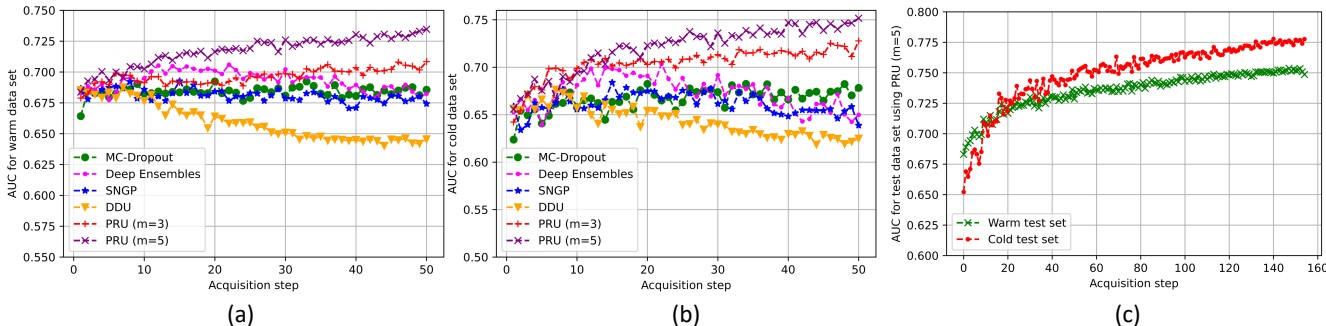

**Figure 7: Active learning setup for the MovieLens-1M dataset. (a) and (b) denotes the AUC after each acquisition step of 1k samples on warm and cold test dataset. (c) AUC improvement using PRU (m=5) for 150 acquisition steps**

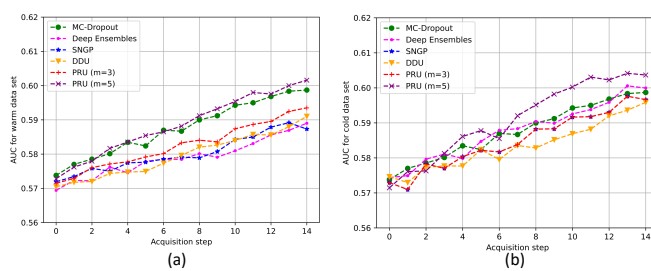

**Figure 8: Active learning setup for the Taobao Ad dataset. (a) and (b) denotes the AUC after each acquisition step of 50k samples on warm and cold test dataset.**

on their frequency out of the total 139 categories, resulting in 1% of the total samples, denoted as Category_id OOD set. We don't train the model on these categories and use them as the OOD test set. We sample an equal number of samples from the rest of the data and use it as the IID test set. We then train the model on the remaining training data using a 90-10 train-val split.

In the second setting, we add noise ($\eta$) to the dense embedding ($e$) for 25% of the OOD test set. This results in more confident scores and maps the OOD samples away from the overall data distribution. Note that this is not an ideal setting, as the model might not experience this type of data distribution in the future. However,

we obtain the OOD test samples that are mapped away from the data distribution. This second setting contains both realistic OOD samples, which the model has not seen and mapped mainly to the overlapping region, and also unrealistic OOD samples that are mapped away from the data distribution.

We plot the feature distribution of OOD/IID test set for Genre#1 MovieLens-1M dataset in Fig 6. The feature distribution for the first scenario is shown in Fig 6 (a,b). Since the model is not trained on OOD samples, we observe that OOD samples get mapped to the class overlapping region, and the model is not confident about the scores for the OOD samples compared to the IID samples. The feature distribution for the second scenario is shown in Fig 6 (c,d). We observe that the majority of the noisy embeddings ($e + \eta$) get mapped away from the actual data distribution and obtain highly confident scores from the model.

We use Wide&Deep as the backbone network and report the AUROC performance for the OOD/IID detection in Table 2. For both settings, we observe that PRU is able to obtain better OOD detection performance compared to the other baseline methods. MC-Dropout and Deep ensemble suffer more in the presence of noisy embeddings, as they are not able to detect OOD samples with highly confident prediction scores. On the other hand, DDU and SNGP performance improve in the presence of noisy embeddings. However, the performance of DDU suffers mainly for the first setting, as DDU flags high confidence prediction IID samples as OOD because high confidence prediction samples lie in the low-density

region for the recommendation system. PRU is able to handle both the cases and thereby improving the performance of OOD/IID detection in both the settings.

## 5.3 Active Learning

Data annotation is an expensive problem for deep learning models. The goal of active learning is to select a small portion of data for labelling from large pool of unlabelled data such that the model constructed with the labeled data has the optimal performance. The key idea behind active learning is that a machine learning algorithm can achieve greater accuracy with fewer training labels if it is allowed to choose the data from which it learns. In uncertainty based active learning, the learner queries samples from the unlabelled set which is least confident (high uncertainty samples), presumably because such labels contain the most information about the downstream task. We follow the active learning setup in the recommendation setting, where at each acquisition step, top N unlabelled samples are picked based on uncertainty and the model is trained on the new generated training set. We start with an initial model trained with 10% of the randomly sampled training data and at each acquisition step push top N uncertain samples from the unlabelled set to the training data. We run it for all uncertainty techniques and report the results for MovieLens-1M and Taobao Ad dataset defined in section 5.2.1.

We divide the dataset into two groups, warm and cold based on their frequency. We use the same terminology and processing used in [32], to define the Warm items (The items whose number of labeled instances is larger than a threshold K). We use K of 200 and 2000 for Movielens-1M and Taobao Ad data. Cold item samples are sorted by timestamp and divided in four equal groups. We use the last (fourth) cold data set sorted based on timestamp as the cold test data set. We sample 10% of the Warm item data as warm test dataset. We use rest of the data as the training data pool. We use Wide&Deep as the backbone model and report the results on the warm and cold dataset. We use N=1k samples for MovieLens-1M and N=50k samples for Taobao Ad in each acquisition step which corresponds to 0.13% and 2% of the unlabelled pool of training data. We run 50 acquisition steps for MovieLens-1M and 15 acquisition steps for Taobao Ad (since each acquisition step of Taobao Ad is expensive) and report the AUC after each acquisition step in Fig 7 and 8. PRU clearly outperforms the baseline uncertainty algorithms. MC-dropout is competitive for the Taobao Ad warm test dataset but require 10 times inference time at each acquisition step. Also, note that we have to train 5 ensemble model at each acquisition step to obtain 5-ensemble uncertainty.

Further the gains are higher for the cold test data set. We plot the AUC improvement performance of PRU on warm and cold test set for extended 150 acquisition steps in Fig 7c. We observe that PRU achieve better performance for the cold test set as uncertainty can be higher for the cold samples as compared to the warm samples in the unlabelled training data. Also we obtain 95% of the baseline model performance (trained with all training data) with 96 acquisition step in MovieLens-1M that is around 13% of the unlabelled pool of data.

## 6 CONCLUSION

In this work, we proposed PRU (Predictive Relevance Uncertainty) for recommendation system. We showed that existing uncertainty estimation techniques suffers for recommendation problems because of the class overlap and class imbalance. We defined the notion of uncertainty for a sample as a distance from the predictive relevance sample of the training data. We showed that the proposed framework is able to correctly define uncertainty for the OOD region and the class overlapping region. We showed the efficacy of the proposed framework in the selective prediction and the downstream task of OOD detection and active learning.

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
