# OpenReview forum: "Predictive Relevance Uncertainty for Recommendation Systems"
_ACM.org/TheWebConf/2024/Conference — TheWebConf24 Oral_

### Official Review · Reviewer_qhRW · 2023-11-17

**Novelty:** 6
**Technical Quality:** 5

**Review:**

The study focus on the uncertainty estimation techniques for recommendation. The authors find the SOTA uncertainty estimation techniques do not work for recommendation problems, and define the uncertainty as a distance from the predictive relevance sample of the training data. Based on the above analysis, the study proposes predictive relevance uncertainty (PRU) to quantify uncertainty for CTR prediction models, then evaluate PRU on three tasks. The experimental results on three datasets prove the effectiveness of PRU.



Advantages:
1.	Uncertainty estimation for recommendation is an important research question.
2.	The authors evaluate the SOTA methods for recommendation tasks, and analyze the special issues in recommendation data sets: overlap and class imbalance.
3.	The authors propose PRU approach to correctly define uncertainty for recommendation task, and evaluate its effectiveness on 3 tasks.

Improvement:

I suggest the authors consider the differences between recommendation task and other machine learning tasks such as image recognition. Where the uncertain come from? In my opinion, people are complex, and their requirements are easily affected by context, so more deeply analysis are needed.

**Questions:**

I suggest the authors consider the differences between recommendation task and other machine learning tasks such as image recognition. Where the uncertain come from? In my opinion, people are complex, and their requirements are easily affected by context, so more deeply analysis are needed.

**Reviewer Confidence:**

3: The reviewer is confident but not certain that the evaluation is correct

**Scope:**

4: The work is relevant to the Web and to the track, and is of broad interest to the community

---

### Official Review · Reviewer_egca · 2023-11-17

**Novelty:** 4
**Technical Quality:** 4

**Review:**

This paper presents an innovative approach to addressing the uncertainty in Click-through Rate (CTR) models, which are pivotal in recommendation systems used across various digital platforms. The authors introduce a novel concept, Predictive Relevance Uncertainty (PRU), designed to better capture the unique predictive uncertainties inherent in recommendation systems, which are not adequately addressed by traditional uncertainty models in classification problems.

The paper demonstrates the practical application of PRU in selective prediction and its utility in tasks such as Out-Of-Distribution (OOD) detection and active learning, while maintaining the efficiency of a single pass deterministic model. The approach appears promising, particularly in enhancing the robustness and reliability of recommendation systems.

However, there are several areas where the manuscript could be strengthened:

1. The introduction lacks adequate literature support for the role of uncertainty estimates in recommendation systems (Lines 90-105). It would be beneficial to provide references to existing work in this area to contextualize the study and establish its relevance.

2. In Figure 2, where the authors visualize data from Avazu and MovieLens, the methods used for this visualization are not clearly described. Elaborating on these methods would enhance the clarity and reproducibility of the results.

3. The choice to use simulated datasets in Section 3, instead of real recommendation data, raises questions about the applicability of the findings to real-world scenarios. It would be useful to provide justification for this choice or consider incorporating real data to validate the findings.

4. The experimental section relies on a limited range of backbone models (DeepFM and Wide&Deep), which are relatively older. Expanding the range of models tested could provide a more comprehensive validation of the PRU approach and its effectiveness across different model architectures.

Overall, while the paper addresses a significant gap in the field of recommendation systems, enhancing its empirical support and broadening the scope of its experimental validation would substantially improve its contribution to the field.

**Questions:**

1. The introduction lacks adequate literature support for the role of uncertainty estimates in recommendation systems (Lines 90-105). It would be beneficial to provide references to existing work in this area to contextualize the study and establish its relevance.

2. In Figure 2, where the authors visualize data from Avazu and MovieLens, the methods used for this visualization are not clearly described. Elaborating on these methods would enhance the clarity and reproducibility of the results.

3. The choice to use simulated datasets in Section 3, instead of real recommendation data, raises questions about the applicability of the findings to real-world scenarios. It would be useful to provide justification for this choice or consider incorporating real data to validate the findings.

4. The experimental section relies on a limited range of backbone models (DeepFM and Wide&Deep), which are relatively older. Expanding the range of models tested could provide a more comprehensive validation of the PRU approach and its effectiveness across different model architectures.

**Reviewer Confidence:**

2: The reviewer is willing to defend the evaluation, but it is likely that the reviewer did not understand parts of the paper

**Scope:**

3: The work is somewhat relevant to the Web and to the track, and is of narrow interest to a sub-community

---

### Official Review · Reviewer_E8bv · 2023-11-26

**Novelty:** 4
**Technical Quality:** 4

**Review:**

The paper proposes Predictive Relevance Uncertainty (PRU) for improving the accuracy of click-through rate (CTR) predictions in recommendation systems. PRU measures how far the new data is from the known and reliable data points. It was tested for effectiveness in various scenarios by using two datasets. The results showed the effectiveness in making recommendation systems more accurate and reliable.

**Questions:**

1.  The paper points out the challenges in CTR prediction from the viewpoint of infrequent occurrence of positive samples, leading to inadequate training data for models with large  number of parameters. This situation results in variability in predictions and difficulties in managing dynamic user behaviors, new customers, and external events. Although the paper proposed Predictive Relevance Uncertainty (PRU) as a solution, I am not fully understand how PRU effectively address these the challenges of class imbalance and overlap, particularly in highly dynamic recommendation scenarios.
2.  PRU involves identifying training samples with significant predictive relevance, fitting a density estimator on these samples' regularized feature space, and then estimating the uncertainty of a test sample accordingly. It would better if the author could give more comprehensive discussion on PRU's computational complexity and feasibility in real-world, large-scale recommendation systems.
3.  The experiments involves a limited range of backbones (e.g., using DeepFM and Wide&Deep). It is better to test the proposed method with more state of the art models as backbones.
4.  The authors mention a number of applications in the introduction section. However, these claims are not tested on in the experiments.

**Reviewer Confidence:**

3: The reviewer is confident but not certain that the evaluation is correct

**Scope:**

3: The work is somewhat relevant to the Web and to the track, and is of narrow interest to a sub-community

---

### Official Review · Reviewer_gNB8 · 2023-11-28

**Novelty:** 5
**Technical Quality:** 4

**Review:**

The issue of relevance estimation is an interesting and useful one, with many possible applications, so the paper takles an interesting problem.

However I have identified severai weaknesses:
First, I found the paper difficult to follow.

Figure 2 shows the "feature distribution" but it is not explained which features are those and how this plot is generated, hence it is not clear what it represents.

Figure 3 with the discussion on the simulation of the behaviour in recommender systems is not very clear to me, the example seems contrived and its parameters seem chosen arbitrarily. I would ask the authors to clarify this since it is used to describe the specific scenario of recommendation.

In the experimental evaluation the preprocessing and data split are not described, instead a reference is provided. I would recommend that the experimental protocol should be clearly described. Especially considering that the referred paper does not in fact indicate how the data is split and preprocessed but one has to follow a further reference to "AutoInt: Automatic Feature Interaction Learning via Self-Attentive Neural Networks.", yet it is stated the splitted data comes from another paper still "Adaptive factorization network: Learning adaptive-order feature interactions."

- The datasets are rather small.

- It is stated that the embedding dimension is set to 16, that is an anomalously low number in my experience. The hyperparameters listed do not appear to have been optimized, which means that the reported methods are not optimal. While the goal of this paper is not to show that any particular recommender is better than another, without proper hyperparameter optimization the risk is that the underlying recommender will exhibit poor quality and have very few highly reliable predictions, biasing the evaluation.

- There is no information on how the baseline uncertainty quantification methods have been optimized. This means we cannot judge the reliability of the experiment.

- It is not clear why the two parts of the analysis use two different datasets. I understand the lack of category information may have prevented the use fo Avazu, but Taobao could have been added to the first experiment. Furthermore, item features are generally bad at representing user interactions, so I would suggest to first check whether they can be used successfully or not. A sample OOD wrt the features may not be at all considering the collaborative information.

**Questions:**

- How was the data processed and split? Why was Taobao not used for the first experiment?
- Was the statistical significance test corrected for the multiple comparisons you are performing?

**Reviewer Confidence:**

2: The reviewer is willing to defend the evaluation, but it is likely that the reviewer did not understand parts of the paper

**Scope:**

3: The work is somewhat relevant to the Web and to the track, and is of narrow interest to a sub-community

---

### Decision · Program_Chairs · 2024-01-22

**Decision:**

Accept (Oral)

**Comment:**

This paper addresses a crucial yet often overlooked issue in click-through rate (CTR) prediction: uncertainty quantification. The authors effectively highlight the unique challenges of uncertainty quantification in CTR prediction, as evidenced by their experimental findings. This aspect is particularly intriguing as it reveals the inadequacy of traditional methods from general classification when applied to CTR prediction. These insights are not only fresh but also highly relevant to the field.

 Furthermore, the authors introduce an innovative approach for quantifying uncertainty in CTR prediction. The extensive evaluation presented in the paper convincingly demonstrates the efficacy of this approach in enhancing CTR predictions. This novel contribution is commendable and adds significant value to the paper.

 I appreciate the paper for pinpointing a key issue in recommender systems that has been underrepresented in current literature. The reviewers have brought up several points for improvement, which should be incorporated in the final version of this paper. In addition, I would like to see the authors share their code for the proposed method and its evaluations.